# Introduction of a New Parameter to Quantify the Fatigue Damage in Asphalt Mastics and Asphalt Binder

**Mohit Chaudhary [1], Nikhil Saboo [2] and Ankit Gupta [1,*]**

1 Department of Civil Engineering, Indian Institute of Technology, Banaras Hindu University, Varanasi 221005, India; mohitchaudhary.rs.civ18@iitbhu.ac.in
2 Department of Civil Engineering, Indian Institute of Technology Roorkee, Roorkee 247667, India; nikhil.saboo@ce.iitr.ac.in
* Correspondence: ankit.civ@iitbhu.ac.in

**Abstract:** This study involves the quantification of fatigue damage in asphalt materials by introducing a new fatigue damage parameter denoted as the F parameter. One waste filler, i.e., red mud and an asphalt binder were chosen to blend the asphalt mastics at three filler contents of 10, 20, and 30% respectively with respect to the volume of binder and tested at temperatures of 5, 15, and 25 °C. The proposed parameter incorporates the effect of both peak shear stress as well as the failure strain, and hence, can better represent the fatigue damage. A lower value of F is recommended for a better fatigue resistant material. The F parameter was found increasing with the increment in filler content, which signifies higher degree of damage with a high level of stiffening. On the other hand, it consistently decreased with the increment in temperature. The behavior of the materials under the action of increasing shear strain was clearly justified by using the F parameter corresponding to different filler contents and the testing temperatures. In addition to that, the observations from the F parameter were also complemented by the fatigue diagrams. Hence, the proposed parameter is envisaged to be a promising fatigue damage indicator in future works.

**Keywords:** fatigue; asphalt mastic; red mud; linear amplitude sweep; viscoelastic continuum damage





## 1. Introduction

The repeated cyclic loading as a result of moving vehicles generates cracking in the aged asphalt pavements. This distress is commonly known as fatigue cracking (or alligator cracking) and generally occurs at the intermediate temperature range of 10 to 30 °C [1]. The composite of mineral filler (<0.075 mm size) and asphalt binder, i.e., asphalt mastic has been acknowledged as the real binder that imparts the adhesion between the aggregates in the asphalt mix [2,3]. In other words, the asphalt mastics can better represent the asphalt mixture performance in comparison to the neat binder. Due to this reason, many researchers have conducted rheological performance tests on the asphalt mastics, rather than neat asphalt binder to evaluate more accurate information about the performance of asphalt mixture [4–7]. The fatigue testing of asphalt mastics is done using the same testing device and the test protocols, and there are no separate standards dedicated for asphalt mastics.

The most conventional fatigue based parameter is the $|G^*|*Sin\delta$, which is measured in the Linear Viscoelastic (LVE) range using an oscillatory test developed during the Strategic Highway Research Program (SHRP) in the early 1990s [8]. The parameter is based on the minimum dissipated energy and involves the basic rheological responses such as complex shear modulus ($|G^*|$) and phase shift angle ($\delta$) [9]. The fatigue parameter is calculated in the LVE domain, which is not in accordance with the actual field conditions where the binders are subjected to Non-Linear Viscoelastic (NLVE) conditions, hence, it is not considered a reliable parameter [10].

The time sweep test developed during the National Cooperative for Highway Research Program (NCHRP) 9–10 is another fatigue test method that employs the repeated sinusoidal loading to the asphalt sample at a constant shear strain amplitude, frequency, and testing temperature [11,12]. The accuracy is very high as the input parameters corresponding to the actual field conditions can be simulated. The major shortcoming of longer testing time makes it a challenging test procedure, especially at low strain rates. Hence, a surrogate test method developed by Johnson et al. [13] came into existence, known as Linear Amplitude Sweep (LAS) test. The test is completed in a very short time period owing to the accelerated damage developed in the testing specimen by linearly increasing load amplitudes.

Though there are various aforementioned test methods that can be employed to generate the fatigue damage in the asphalt samples, the real challenge is to accurately predict the failure point within the specimen. Conventionally, the failure point is characterized by the traditional approach, in which, the point corresponding to the 50% reduction in the initial $|G^*|$ is considered as the failure of the test specimen in fatigue [14]. Obviously, this definition is empirical, and hence may not represent the true failure. Likewise, the researchers experimented and suggested other failure criteria, such as peak in phase angle [15,16], Dissipated Energy Ratio [1,17], and Ratio of Dissipated Energy Change [18,19] approach, pseudo strain energy approach [20,21], etc. Due to the drawbacks associated with each of the fatigue failure definitions, there is no universal fatigue damage indicator available to date. Hence, the authors in this study aimed to provide a new fatigue failure parameter based on the fundamental stress–strain behavior of the material in a typical LAS test. Based on the research gap, the following objectives are outlined:

(I)　To introduce a new fatigue parameter based on the typical LAS test for the determination of accurate fatigue failure points in the asphalt mastics and asphalt binder.

(II)　To investigate the effect of filler content and the testing temperature on the fatigue behavior of the asphalt materials.

Figure 1 presents the flow chart for the research outline of the study.

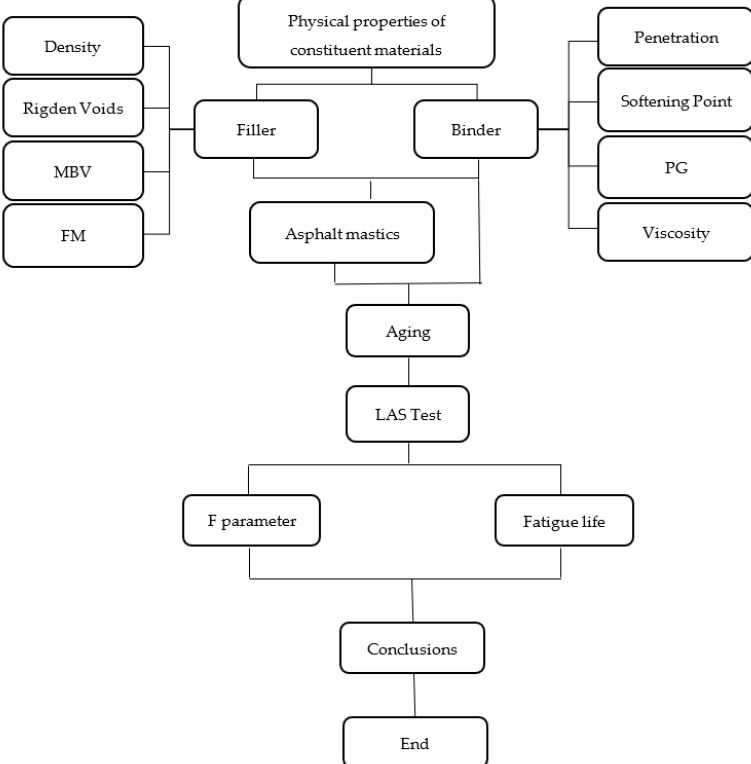

**Figure 1.** Flow chart for the research outline of the study.

## 2. Materials

With the aim of utilizing the waste materials as an alternative for the conventional mineral fillers, red mud (RM) was chosen as the filler. It is an industrial by-product generated during the production of alumina from bauxite ore through Bayer's process [22]. The physical properties of the RM are shown in Table 1. The grain size distribution of the filler is illustrated in Figure 2. The paving asphalt binder of viscosity grading (VG-30 or AC-30) was incorporated in this study. The basic properties of the asphalt binder are given in Table 2.

**Table 1.** Physical properties of filler.

| Property | Value |
|---|---|
| Particle density (g/cm$^3$) | 3.22 |
| Void content by Rigden (%) | 45.7 |
| Methylene blue value (g/kg) | 2.5 |
| Fineness modulus | 4.32 |
| Diameter $d_{60}$ (µm) | 41 |
| Diameter $d_{10}$ (µm) | 1.11 |

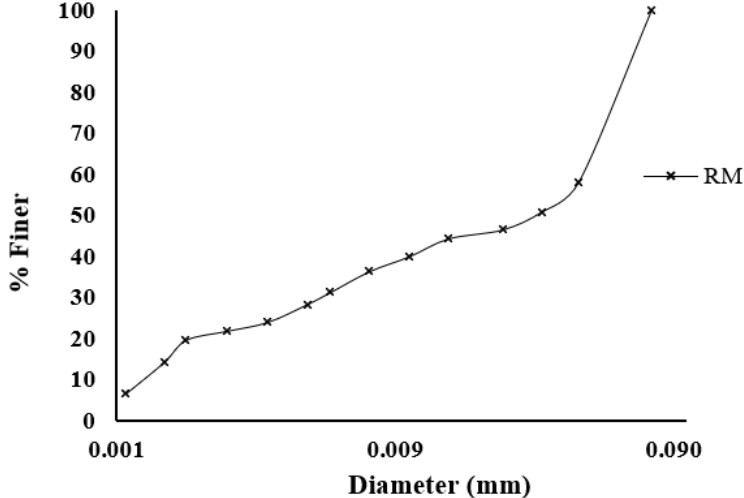

**Figure 2.** Particle size distribution of filler.

**Table 2.** Basic properties of binder.

| Property | Value |
|---|---|
| Penetration, dmm | 62 |
| Softening Point, °C | 48 |
| Viscosity at 60 °C, Poises | 2704 |
| High Temperature PG Grade | PG 70-22 |
| Continuous Upper PG, °C | 74.30 |

## 3. Experimental Investigations

### 3.1. Physical Properties

The mineral filler being an integral part of the asphalt mastics plays a vital role in its performance. A plethora of research is available that states that the filler characteristics such as particle size, shape, composition, etc. are the dominant factors affecting the behavior of the asphalt mastics [23,24]. In order to understand the effect, the filler properties were evaluated by a set of characterization tests. The standard pycnometer method conforming to ASTM D 854 [25] was employed to evaluate the specific gravity of fillers. The grain size distribution of the RM was obtained with the help of hydrometer analysis [26]. It provides the grain size distribution curve by relating the specific gravity of the filler suspension

and the mass of solids present. The grain size distribution provides only the qualitative outlook about the sizes of the particles through the distribution curve. The mathematical quantification of the size of the particles can be done by a relative fineness parameter known as the Fineness modulus (FM). The cumulative summation of the percentages of filler coarser than 75, 50, 30, 20, 10, 5, 3, and 1 μm divided by 100 gives the value of FM. The lower magnitude of FM represents the fine particle sizes. The intergranular porosity of the fillers was evaluated with the Rigden Voids (RV) test, which determines the volume of air-filled voids in a dry compacted bed of fines [27]. The deleterious materials present in the fillers such as harmful clays (smectite group), organic matter, iron hydroxides, etc. can significantly alter the test results and therefore should be present in allowable quantity. It was quantified by the Methylene Blue Value (MBV) test [28]. The detailed description on the filler characterization tests can be found elsewhere [29].

### 3.2. Fabrication of Mastics

The asphalt mastics were fabricated in the laboratory at three filler contents ($\varphi$), which is the ratio of the amount of filler with respect to the amount of binder in the mastic. The ratio of filler and the binder was taken on the volume basis due to the high accuracy as compared to the weight method, as different fillers may occupy different volumes within the mastic owing to their variable specific gravities and hence may cause non-uniformity. The three chosen filler contents were 10, 20, and 30%. The required amount of filler and binder corresponding to each $\varphi$ were weighed on the separate pan. The binder was heated at 180 °C in the draft oven for 10 min so that it became fluid enough to mix with filler [29]. The pan containing the filler was also kept in the oven at the same temperature for a 1 h time duration. The binder pan was then taken out from the oven and placed on the hot plate. The filler was added to the binder slowly under continuous agitation by a manually operated mixer for 10 min. The mixing and stirring were done continuously to obtain the homogenous mixture of filler and the binder. After mixing, the container was placed in the freezer to prevent the filler particles from settling at the bottom due to the high temperature of the binder.

### 3.3. Ageing Procedure

The short-term ageing (STA) simulates the ageing during mixing and compaction of the asphalt mixture at the time of production, whereas the long-term ageing (LTA) was done to create the field conditions corresponding to several years of ageing. Generally, rolling thin film oven test (RTFOT) [30–33] and Pressure ageing vessel (PAV) [34] are used for STA and LTA, respectively. This study used another ageing method known as the Universal Simple Ageing Test (USAT) method introduced by Western Research Institute (WRI) [35]. In this method, a conventional draft oven was used for both STA and LTA. The mastic samples were spread in the flat plate with the help of a spatula to a very thin layer thickness. The plate was then kept in the draft oven for a period of 50 min at a temperature 150 °C for STA. The STA samples in the plate were again heated in the oven for 40 h at a temperature of 100 °C. After the ageing, the materials were cooled and scrapped with a scrapping tool and then stored in separate cans for further testing.

### 3.4. Rheological Testing

The asphalt samples were subjected to rheological testing with the help of an Anton Paar MCR 102 dynamic shear rheometer (DSR). The parallel geometry including 8 mm sample diameter and 2 mm height was used for the testing. The samples were fabricated using the silicon mold method in which the hot mastic was poured and cooled in the mold. The samples were then extracted from the silicon mold in the form of cylindrical pellets. The sample was placed between the parallel plates with the lower being stationary and the upper plate applying the oscillatory loading with the use of a spindle. The gap was then fixed to 2 mm by lowering the spindle and the sample was sandwiched between them. The excess sample was trimmed from the periphery by the hot trimming tool. The equilibrium

time of 10 min at the specific testing temperature was maintained before testing the sample to allow the uniform temperature within the sample. All the tests were done twice and the mean of both replicates is presented in the study. The variation between the replicates was kept well within 10% and a third replicate was tested if the results cross the 10% variation. Figure 3 displays the asphalt specimen undergoing the LAS test using the DSR.

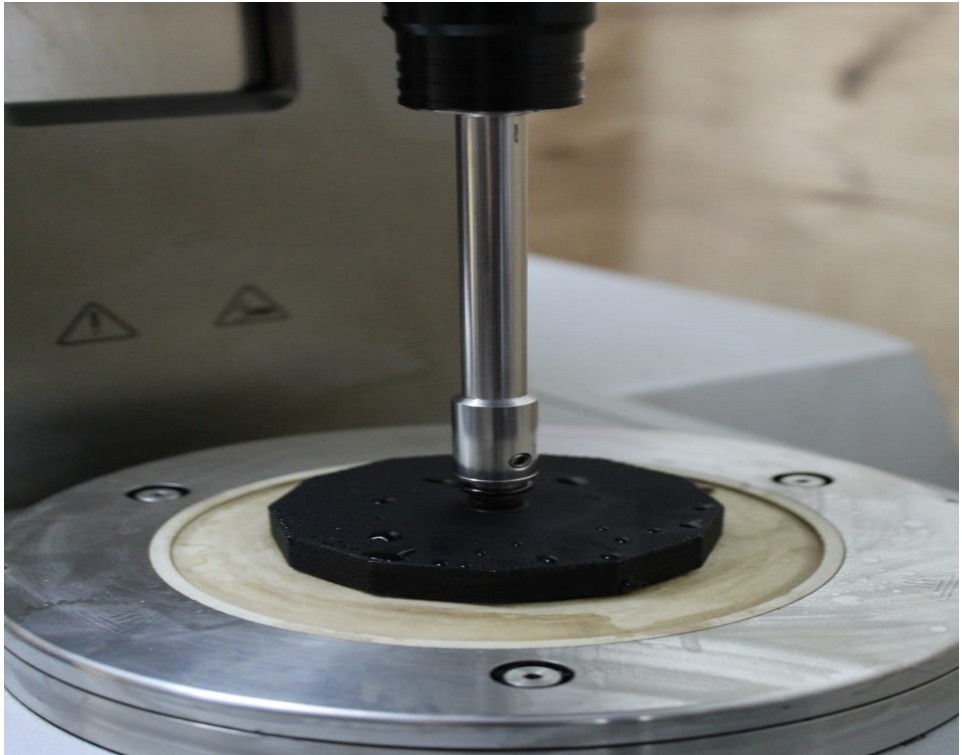

**Figure 3.** Specimen under experiment.

LAS Test

The accelerated fatigue test method, i.e., LAS test, is a two-step test process: frequency sweep (FS) test followed by an amplitude sweep test [36]. The undamaged and damaged characteristics of the material can be obtained through a single test. The primary aim of conducting the FS test as a part of the LAS test is to obtain the undamaged material property denoted by $\alpha$. It is then followed by an amplitude sweep from 0.1 to 30% at a fixed frequency of 10 Hz. Due to such high strain magnitudes, the material goes from LVE to NLVE domain in a short time with the linearly increasing shear strain. The damaged properties of the material are then utilized to obtain the fatigue life of the material at any strain amplitude using the viscoelastic continuum damage (VECD) method. The VECD model helps in obtaining the relationship between the number of cycles to failure ($N_f$) or fatigue life and the applied strain.

*3.5. F Parameter*

The stress within the material increases with the increase in applied strain and reaches the maximum value. This maximum value is known as peak shear stress and the corresponding strain is called failure strain. Due to this, the typical stress–strain curve shows a rapid rise, reaches its peak, and starts decreasing with the onset of damage. If the material remains undamaged with the increasing damage level, then the stress–strain becomes an ever increasing straight line, and the peak, as well as the latter part of the curve, is not observed. The corresponding stress in that condition is known as pseudo stress [37]. The difference between the peak stress in the actual stress–strain curve and the corresponding stress in the pseudo stress–strain curve is the actual damage that has been done in the

sample by the increasing shear strain amplitude. The authors, in this study, focused on quantifying this difference to determine the actual damage in the asphalt sample.

In addition to that, the failure strain is also equally important as it determines the strain tolerance of the sample. There can be some cases in which a material has very high peak stress, but also has a high failure strain due to which assessing the damage in the material based on peak shear stress alone may not give accurate results. Therefore, it is better to have a single parameter that can predict the damage in the material based on both peak shear stress and the failure strain. Hence, the F parameter in the form of a mathematical index is used in the study to quantify the fatigue damage in the asphalt materials.

Mathematically, it can be expressed as the ratio of the percentage difference between peak shear stress and the corresponding peak pseudo stress ($\tau_D$) in the material divided by the failure strain. The higher peak stress signifies that the material is stressed to a high magnitude as a result of stiffening in the asphalt mastic, which corresponds to the shorter fatigue life. On the other hand, the high failure strain point towards the higher strain tolerance of the material and hence yields better fatigue life. Theoretically, a lower value of peak stress and higher failure strain indicates high fatigue performance. Therefore, a lower value of the F parameter is recommended for superior fatigue performance.

$$F = \frac{\tau_D}{\gamma_F} \qquad (1)$$

$$\tau_D = \frac{\tau_p - \tau_A}{\tau_p} * 100 \qquad (2)$$

where $\tau_p$ is the peak pseudo stress (Pa), $\tau_A$ is the peak shear stress (Pa), and $\gamma_F$ is the failure strain (%) in the asphalt sample. Figure 4 shows the actual stress–strain curve (lower) and the pseudo stress–strain curve (upper) for an arbitrary sample. The peak stress and the peak pseudo stress are marked with the arrows in the left at the ordinate axis. The failure strain is shown by a downward arrow at the abscissa axis. The difference between the peaks of both the curves shown by a double-pointed arrow is the actual damage occurring in the sample.

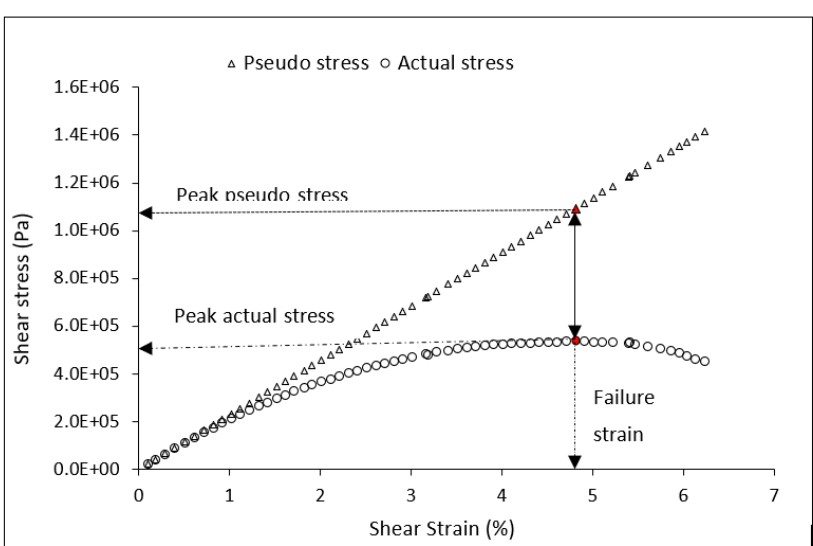

**Figure 4.** Variation of actual stress and pseudo stress as a function of shear strain.

## 4. Results and Discussion

### 4.1. F Diagrams

Figure 5a shows the variation of the F parameter with the change in temperature for neat binder and asphalt mastics. It was observed that the value of the F parameter continuously decreased at higher temperatures. This behavior was true for neat binder,

10%, and 20% filler content except for 30% filler content, where it first increased at 15 °C and then decreased at 25 °C. At higher temperatures, the binder became soft and hence can resist higher strain amplitude before reaching the maximum stress value. In view of this, the fatigue resistance of the binder and mastics at 10 and 20% φ increased consistently at higher temperatures. On the other hand, the volume occupied by the filler particles in the asphalt matrix at 30% φ was very high, due to which the over stiffening effect took place, and hence the effect of temperature increment could not be observed at 15 °C. On further elevating the temperature, the value of F again decreased and hence it started portraying a similar behavior as of other materials at 25 °C. In other words, the temperature of 15 °C was not high enough to diminish the effect of high filler content, i.e., 30%, but the temperature effect became dominant at 25 °C. It may also be possible that at 15 °C, the binder became soft and the mobility of the filler increased, which caused the accumulation of the particles at some places within the mastic, due to which, the stiffness became very high and consequently the F value increased unexpectedly. But at 25 °C, the binder was hot enough to allow the uniform dispersion of the particles and improve the strain resistivity resulting in a lower F value.

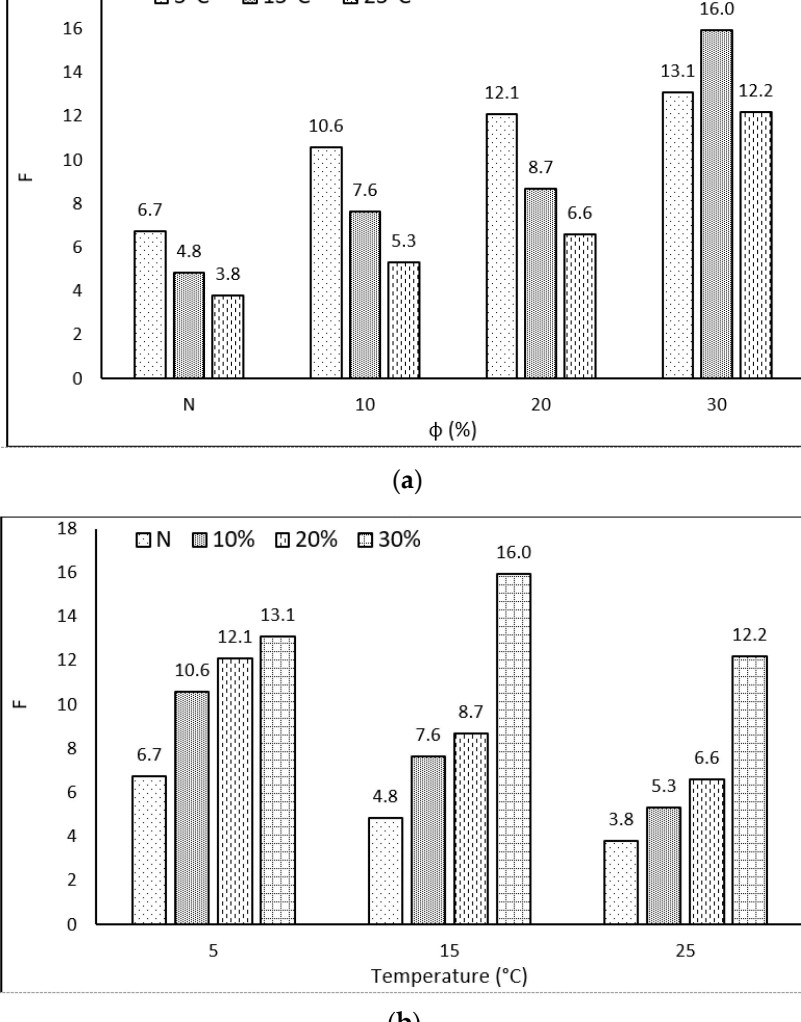

**Figure 5.** Variation in F parameter with the change in (**a**) testing temperature and (**b**) filler content.

From Figure 5b, it is clear that the higher doses of filler exhibit higher F values. Moreover, the F values of the neat binder were always lower than that of all the mastics. A similar type of behavior was observed at all testing temperatures. The addition of

filler to the neat binder caused a stiffening effect in the binder, due to which, the peak stress increased and also the material reaches the peak value rapidly, i.e., at lower strain amplitudes. As a result of this, the F parameter was pushed to a higher value in comparison to the neat binder. Similarly, with the further increment in filler content, the stiffness of the mastic consistently increased and hence displayed higher F values at high filler contents. Similar values of the F parameter were observed at 5 °C, corresponding to all filler contents. On increasing the temperature to 15 °C, the F values for 10 and 20% filler content were similar, but the value corresponding to 30% filler content was considerably higher. This observation holds true for 25 °C temperature also. This implies that the relative stiffening at 30% filler content was very high as compared to 10 and 20% filler content. The excessive high values at 30% can be the over stiffening caused by filler and it is not recommendable from the fatigue point of view. Therefore, the permissible filler dosage can be kept to 20% in red mud, as observed from the F parameter.

To highlight the importance of failure strain in the F parameter, the evolution of $\tau_D$ values as a function of temperature and the filler content has been shown in Figure 6. It is clear that the change in $\tau_D$ followed a similar trend as of F with the change in filler contents at various temperatures, but it is worth mentioning that the relative difference in the absolute values of $\tau_D$ and F was dissimilar. While the relative difference in $\tau_D$ values at different filler contents was approximately similar, the change in the F parameter from 20 to 30% was massively high, as explained in the previous section.

In addition to that, $\tau_D$ and F displayed a totally different behavior with the change in temperature at different filler contents as there was no uniform pattern observed. For example, $\tau_D$ continuously decreased at 10% φ, increased and then decreased at 20% filler content, consistently increased at 30% φ with the temperature increment. On the other hand, a specific trend was observed in the variation of F as explained in the earlier sections. It is clear that the inclusion of failure strain has totally changed the effect of temperature on the behavior of asphalt mastics. Therefore, it can be concluded that the fatigue characteristics as a result of the stiffening effect of fillers in the asphalt mastics cannot be explained by shear stress alone. This has motivated the authors to include both peak shear stress and failure strain in the fatigue damage indicator, i.e., F parameter. The explanation of fatigue behavior of the asphalt mastics and the asphalt binder by the F parameter has also been verified by the fatigue diagrams in the later sections to check the reliability of the parameter.

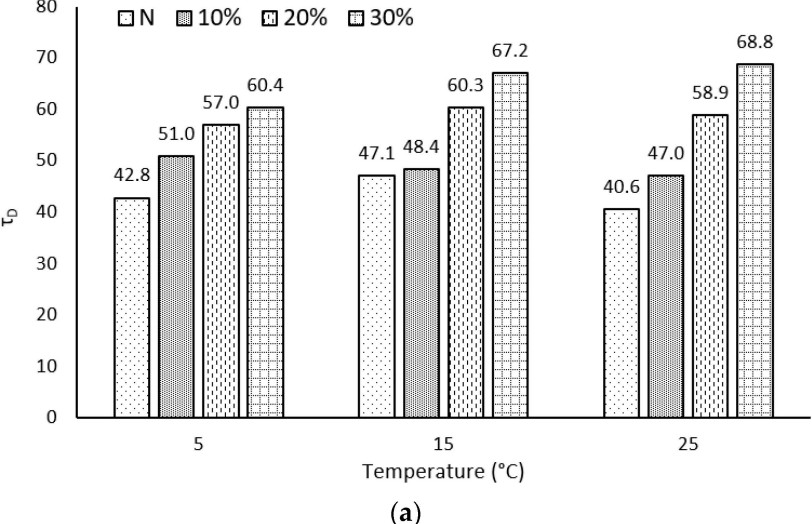

(a)

**Figure 6.** *Cont.*

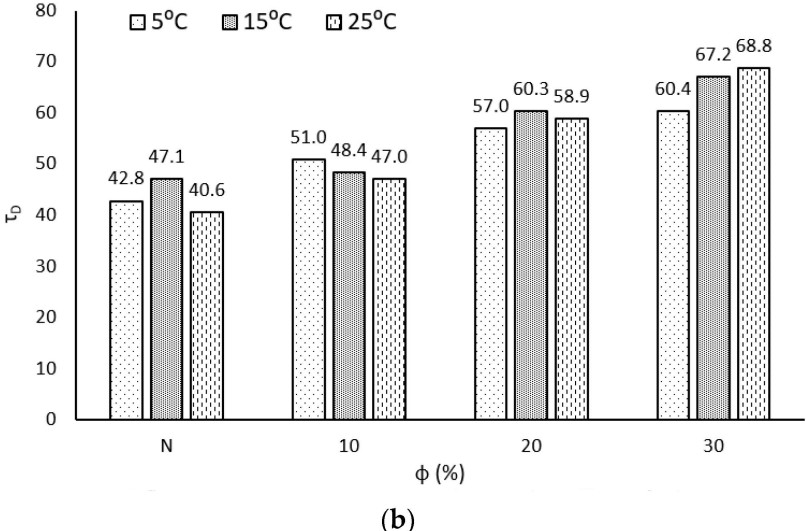

(**b**)

**Figure 6.** Variation in $\tau_D$ with the change in (**a**) filler content and (**b**) testing temperature.

### 4.2. Fatigue Diagrams

Figure 7 shows the variation of fatigue lives of the neat binder as well as asphalt mastic in terms of the number of cycles to failure ($N_f$) as a function of shear strain amplitude. It has been done to assess whether the results obtained from the F parameter results complement the fatigue behavior of the asphalt materials as observed from the fatigue diagrams. The neat binder, mastics with 10%, and 20% filler content showed similar rheological responses to the varying temperatures. The fatigue lives for these asphalt samples were found increasing with the higher temperatures, which is obvious, as the material became less stiff with the increase in temperature and hence yielded higher fatigue lives. It was also observed that the fatigue curves at 15 and 25 °C almost overlapped, signifying similar fatigue lives, whereas the fatigue life at 5 °C was the lowest. On the other hand, the aforementioned observations were not true for 30% filler content, where the Nf decreased at 15 °C and became highest at 25 °C. Similar outcomes were observed from F diagrams as well, as shown in Figure 5.

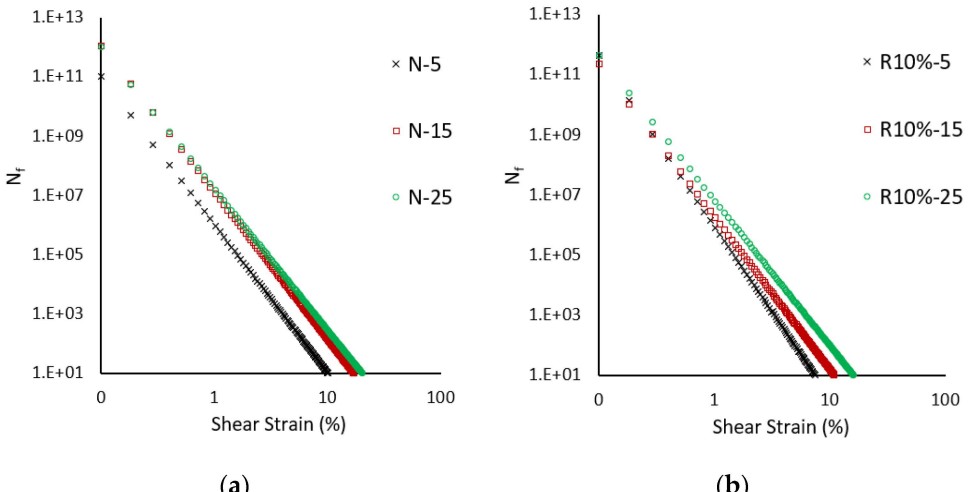

(**a**) (**b**)

**Figure 7.** *Cont.*

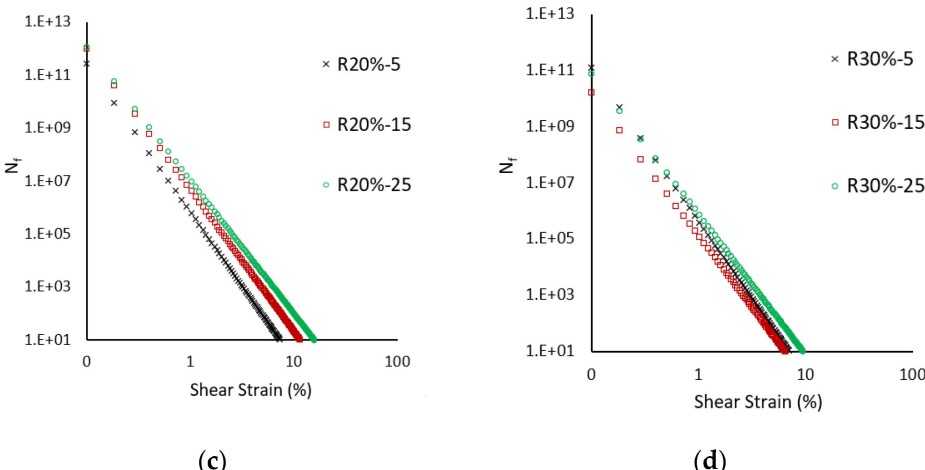

(c)                                                (d)

**Figure 7.** Fatigue life as a function of shear strain at different temperatures for (**a**) neat binder (**b**) 10%, (**c**) 20%, and (**d**) 30% filler content.

Figure 8 shows the effect of filler addition by comparing the fatigue diagrams of the neat binder and all the asphalt mastics at individual testing temperatures. It was observed that the fatigue curves for all the filler contents superimposed on one another representing identical fatigue lives, whereas the neat binder showed the highest fatigue life at 5 °C. On increasing the temperature to 15 °C, $N_f$ corresponding to neat binder was found highest followed by 10, 20, and 30% filler contents, respectively. It is also noted that the mastics having 10 and 20% filler content had similar fatigue lives, whereas the $N_f$ at 30% filler content was comparatively very low. On further increasing the temperature to 25 °C, no considerable effect of lower filler contents, i.e., 10 and 20% was noted as evident from similar fatigue curves of neat binder, 10%, and 20% filler content. However, the fatigue life at 30% was also significantly less similar to what had been observed at 15 °C temperature.

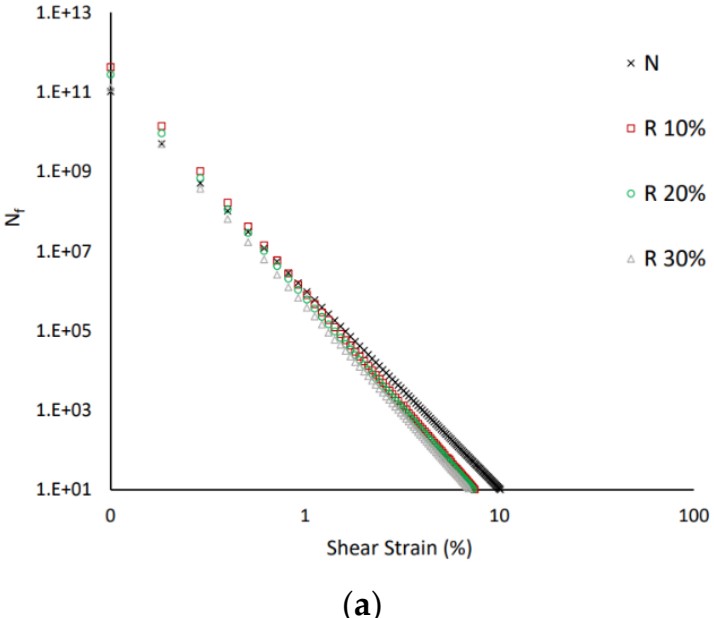

(**a**)

**Figure 8.** *Cont*.

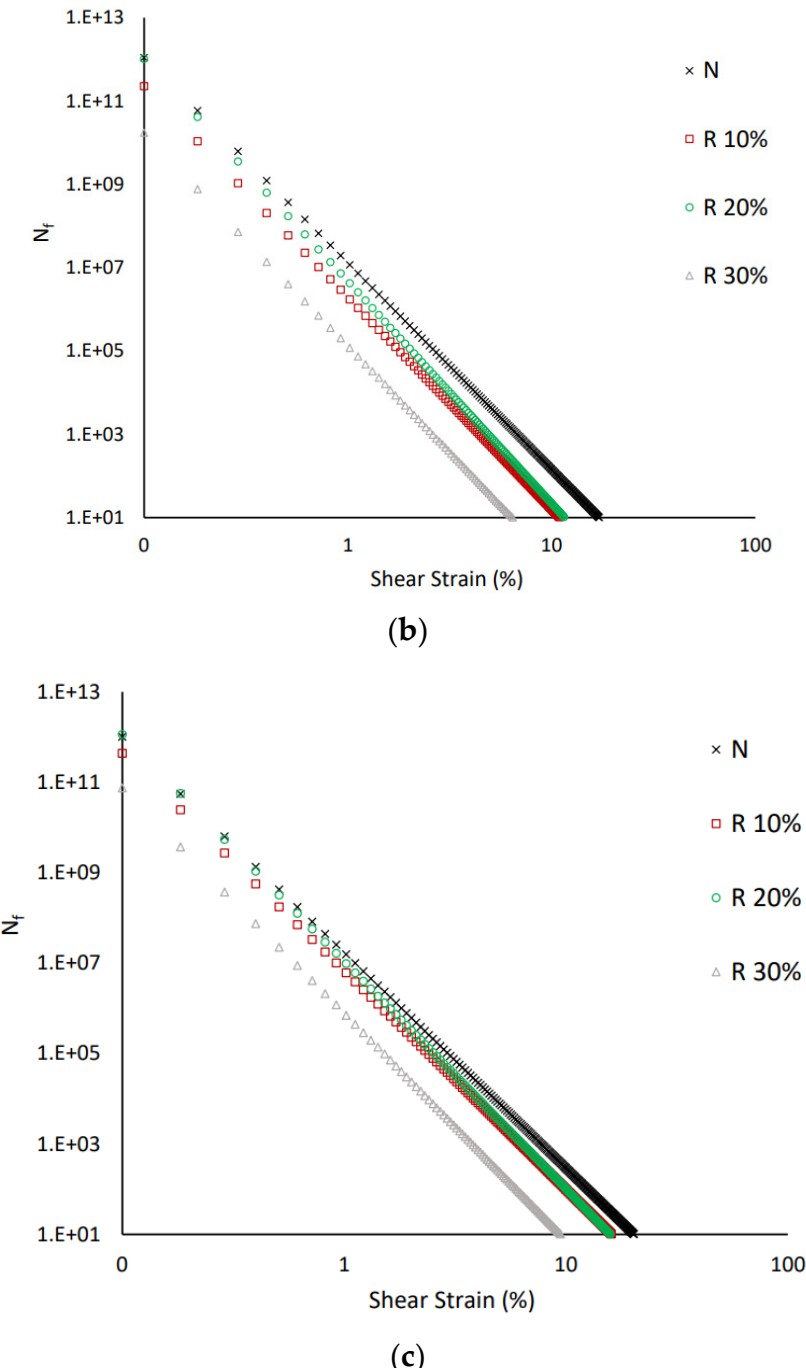

**Figure 8.** Fatigue life as a function of shear strain at (**a**) 5 °C, (**b**) 15 °C, and (**c**) 25 °C.

From Figure 8b,c, it is clear that the fatigue life at 30% filler content was considerably less and the behavior at this filler content was totally different from the remaining filler contents. The drastic decrease in the fatigue life at 30% filler content can be attributed to the over stiffening caused by the filler particles, and hence the filler content can be restricted to 20%.

The aforementioned observations from fatigue curves were totally in accordance with the behavior observed from the F parameters in Figure 5. Hence, it can be concluded that the newly proposed fatigue damage indicator in the form of the F parameter effectively represents the true fatigue behavior of the asphalt materials.

## 5. Conclusions

This study aims to provide a new parameter for the quantification of fatigue damage in the asphalt mastic, as well as the asphalt binder based on the fundamental stress–strain behavior of the material under the action of LAS test. Red mud as a waste filler and a neat binder were incorporated to blend the asphalt mastics at three filler contents and tested at three intermediate temperatures via the LAS test. Due to the unavailability of universal fatigue damage indicator, the F parameter can be used to obtain an accurate interpretation of fatigue damage in the asphalt materials. The specific findings from the study are as follows:

- The typical stress–strain curve from the LAS test can be employed to assess the true fatigue behavior of the asphalt materials. The fatigue behavior cannot be predicted with the peak shear stress alone, but the failure strain should also be incorporated.
- The behavior of the materials signified from the F parameter was also justified from the fatigue curves. Hence, the F parameter was proved to be a good indicator for the fatigue damage in asphalt mastics as well as the asphalt binder.
- The upper limit of filler dosage can be limited to 20% for the red mud as the substantial high values of F parameter at 30% filler content indicated poor fatigue performance owing to the over stiffening effect.

The efficacy of the F parameter can be assessed on a much wider scale with the incorporation of different fillers having varying properties and other binders like polymer modified binder in future studies.

**Author Contributions:** Experiment, data analysis, and manuscript preparation: M.C.; Conceptualization, supervision, and manuscript preparation: N.S.; Supervision, drawing logical inferences from results, and manuscript correction: A.G. All authors have read and agreed to the published version of the manuscript.

**Funding:** This research received no external funding.

**Institutional Review Board Statement:** Not applicable.

**Informed Consent Statement:** Not applicable.

**Data Availability Statement:** Not applicable.

**Acknowledgments:** The authors would like to thank the Indian Institute of Technology (Banaras Hindu University) for facilitating the laboratory study.

**Conflicts of Interest:** The authors declare that they have no conflict of interest.

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
