# Peer review of "Introduction of a New Parameter to Quantify the Fatigue Damage in Asphalt Mastics and Asphalt Binder"

_coatings, doi:10.3390/coatings11070828_

Round 1

Reviewer 1 Report

I would like to congratulate the authors on submitting an interesting work with data of practical use potential.

Author Response

Review Report (Reviewer 1):

I would like to congratulate the authors on submitting an interesting work with data of practical use potential.

Response: The authors would like to thank the reviewer for the appreciation.

Reviewer 2 Report

This study involves the quantification of fatigue damage in asphalt materials by introducing a new fatigue damage parameter denoted as the F parameter on asphalt mastics made up by red mud in comparison with an asphalt binder. The work is interesting and well-structured.

Some suggestions for minor revision are as follows:

  • Line 13: The verb “to fabricate” is not appropriate for mastics. It is suggested to use either "to mix" or "to blend".
  • Lines 13 and 14: Please, specify compared with what the three percentages of filler content have been calculated.
  • Line 37: The authors are invited to include additional research works that deal with asphalt mastics. More details are given in: "Investigating the rheological properties of hot bituminous mastics made up using plastic waste materials as filler"
  • Line 40: With “|G*|.sinδ”  The multiplication sign is missing.
  • Line 44: Please, a research work should be added for supporting the meaning of the |G*| and δ. More details are given in: "Verifying laboratory measurement of the performance of hot asphalt mastics containing plastic waste"
  • Lines 68 to 74: The description of the research work should be improved. It is suggested to add a flow chart to summarize the main steps of the research.
  • Line 79: It is the first time that RM appears in the text. What does it mean?
  • Line 80: Please add the standard adopted for the “hydrometer analysis”.
  • Lines 94-95: Why do the authors talk again about the hydrometric analysis if already anticipated in the previous section? It is suggested to discuss on the grading curves in a only one section.
  • Lines 117: Please explain why the authors have adopted 180°C as blending temperature
  • Line 129: It is recommended to add research papers that deals with RTFOT procedure More details are given in: "Cold Techniques and Materials for Sustainable Pavement Construction and Rehabilitation"
  • Line 139: What is the manufacturing house and model of the rheometer?
  • Line 223 to 226: It would be interesting to compare the mastics with red mud and those with traditional filler.

Author Response

Review Report (Reviewer 2):

This study involves the quantification of fatigue damage in asphalt materials by introducing a new fatigue damage parameter denoted as the F parameter on asphalt mastics made up by red mud in comparison with an asphalt binder. The work is interesting and well-structured.

Response: The authors would like to thank the reviewer for the appreciation.

Some suggestions for minor revision are as follows:

Line 13: The verb “to fabricate” is not appropriate for mastics. It is suggested to use either "to mix" or "to blend".

Response: The verb “to fabricate” has been changed to “to blend” in the manuscript as per reviewer’s suggestion.

Lines 13 and 14: Please, specify compared with what the three percentages of filler content have been calculated.

Response: Thanks! The three filler contents have been calculated with respect to the volume of binder.

Line 37: The authors are invited to include additional research works that deal with asphalt mastics. More details are given in: "Investigating the rheological properties of hot bituminous mastics made up using plastic waste materials as filler"

Response: The additional research works have been added in the manuscript now.

Line 40: With “|G*|.sinδ”  The multiplication sign is missing.

Response: The multiplication sign has been added now.

Line 44: Please, a research work should be added for supporting the meaning of the |G*| and δ. More details are given in: "Verifying laboratory measurement of the performance of hot asphalt mastics containing plastic waste"

Response: The research work has been added in the manuscript now.

Lines 68 to 74: The description of the research work should be improved. It is suggested to add a flow chart to summarize the main steps of the research.

Response: The flow chart has been added now in the manuscript as Figure 1.

Line 79: It is the first time that RM appears in the text. What does it mean?

Response: Many Thanks! The RM stands for Red Mud here. The authors have corrected this error now in the manuscript. The acronym “RM” has been added in the line 81 and hence the term “RM” in line 83 is appropriate now.

Line 80: Please add the standard adopted for the “hydrometer analysis”.

Response: The description of hydrometer analysis has been done in section 3.1 in line 99-100 now as per the reviewer’s suggestion in next comment. The standard for hydrometer analysis has been added there (Reference [26]).

Lines 94-95: Why do the authors talk again about the hydrometric analysis if already anticipated in the previous section? It is suggested to discuss on the grading curves in a only one section.

Response: Many Thanks! The discussion on the hydrometer analysis have been removed now from the section 2. It is now described only in section 3.1 in line 99-100.

Lines 117: Please explain why the authors have adopted 180°C as blending temperature.

Response: The blending of the asphalt mastics has been done by following the similar protocol as per previous work done by Chaudhary et al. [3]. Therefore, a blending temperature of 180°C has been selected to mix the asphalt mastics.  

Line 129: It is recommended to add research papers that deals with RTFOT procedure More details are given in: "Cold Techniques and Materials for Sustainable Pavement Construction and Rehabilitation"

Response: Many Thanks! The research works dealing with RTFOT procedure have been added in the manuscript now.

Line 139: What is the manufacturing house and model of the rheometer?

Response: The Anton Paar MCR 102 Dynamic Shear Rheometer (DSR) has been used for the rheological testing of the asphalt mastics and the asphalt binder.  

Line 223 to 226: It would be interesting to compare the mastics with red mud and those with traditional filler.

Response: Many Thanks! The authors found the idea of comparing the asphalt mastics with red mud filler and traditional filler very interesting as suggested by the reviewer. We will try to incorporate this suggestion in the future study. 

Reviewer 3 Report

The article "Introduction of a new parameter to quantify the fatigue dam-2 age in asphalt mastics and asphalt binder" presents an interesting topic and is presented pretty well however, I have some minor questions and suggestions. 

First, what is the significance of this parameter "F" in comparison to the original pick stress/strain values because it is pretty much based on those two values? 

Second, does this parameter and method works for other binder types as well like rubber and other type of modified binders? I know this article is more for pure binder. Just as example the following paper shows the data for rubberized binder (https://doi.org/10.1016/j.cscm.2019.e00276 ). Do you think you can recommend it for other binder types?

Moreover, I suggest the following improvements,

A photo showing the specimen under experiment will help the paper looks better.

In Civil Engineering, specially in US, AC-30 is more common term. I think it is good if you add it in parenthesis in front of VG-30.

All parameters presented in the equations and figures should have units.  

Author Response

Review Report (Reviewer 3)

The article "Introduction of a new parameter to quantify the fatigue damage in asphalt mastics and asphalt binder" presents an interesting topic and is presented pretty well however, I have some minor questions and suggestions.

First, what is the significance of this parameter "F" in comparison to the original pick stress/strain values because it is pretty much based on those two values?

Response: The authors in this study attempted to quantify the damage in the sample by utilizing both peak stress and failure strain. The difference between the peak stress in the actual stress-strain curve and the corresponding stress in the pseudo stress-strain curve is the actual damage that has been done in the sample by the increasing shear strain. In addition to that, the failure strain is also equally important as it determines the strain tolerance of the sample. Also, it may be possible that a material has very high peak stress and a high failure strain due to which assessing the damage in the material based on peak stress alone may not give accurate results. Therefore, a single parameter having both peak stress and the failure strain values can quantify the actual damage in the sample in a more accurate way and hence the F parameter is introduced in this study. 

Second, does this parameter and method works for other binder types as well like rubber and other type of modified binders? I know this article is more for pure binder. Just as example the following paper shows the data for rubberized binder (https://doi.org/10.1016/j.cscm.2019.e00276 ). Do you think you can recommend it for other binder types?

Response: Thanks! The authors incorporated the asphalt mastics in addition to the pure binder to justify the novelty of the proposed parameter. In addition to that, the mastics were blended at three filler contents (10, 20, and 30 %) and tested at the three intermediate temperatures of 5, 15, and 25 °C to study the efficacy of the F parameter at a wide range of variables.  As per the results, the F parameter was proved to be a good indicator for the fatigue damage in asphalt mastics as well as the pure binder. Therefore, the authors recommend this parameter as a reliable fatigue damage indicator for other types of binders as well including rubber modified binders. Also, the efficacy of using the F parameter to investigate the fatigue damage in asphalt materials incorporating different types of binders can be a promising future study.

Moreover, I suggest the following improvements,

A photo showing the specimen under experiment will help the paper looks better.

Response: Thanks! The photo of the specimen under experiment has been added to the manuscript in the form of Figure 3.

In Civil Engineering, specially in US, AC-30 is more common term. I think it is good if you add it in parenthesis in front of VG-30.

Response: Many Thanks! The suggestion has been incorporated in the manuscript now.      

All parameters presented in the equations and figures should have units. 

Response: Thanks! The missing units in the equations and figures have been updated now.

Reviewer 4 Report

Dear Authors,

Thank you for your well-written manuscript, here will be following minor remarks:

Line 10, 20 and 30% per binder weight? Please add details in description

Aging -> ageing

Short-term, long-term

Please use symbols and not letters to indicate degrees: 50oC -> 50°C (insert -> symbol->°)

Red Mud -> red mud

In conclusions, before bullet points is good idea shortly to include info about what is that F parameter and to underline the novelty of your study.

Author Response

Review Report (Reviewer 4)

Dear Authors,

Thank you for your well-written manuscript, here will be following minor remarks:

Line 10, 20 and 30% per binder weight? Please add details in description

Response: The filler contents were chosen on the volume basis. The study is conducted on the asphalt mastics at three filler contents of 10, 20, and 30 % with respect to the volume of binder. The details have been added in line 13-14 in the manuscript now.  

Aging -> ageing, Short-term, long-term, please use symbols and not letters to indicate degrees: 50oC -> 50°C (insert -> symbol->°), Red Mud -> red mud

Response: Many Thanks! All the aforementioned suggestions have been incorporated in the manuscript now.

In conclusions, before bullet points is good idea shortly to include info about what is that F parameter and to underline the novelty of your study.

Response: Many Thanks! The authors have now added the info about the F parameter and also the novelty of the study in the conclusions section as suggested by the reviewer.